# Sedentariness of College Students Is Negatively Associated with Perceived Neighborhood Greenness at Home, but Not at University

**DOI:** 10.3390/ijerph17010235

**Published:** 2019-12-28

**Authors:** Alexander Karl Ferdinand Loder, Mireille Nicoline Maria van Poppel

**Affiliations:** 1Institute of Sport Science, University of Graz & Staff Department Quality Management, University of Music and Performing Arts Graz, Graz 8010, Austria; 2Institute of Sport Science, University of Graz, Graz 8010, Austria; mireille.van-poppel@uni-graz.at

**Keywords:** neighborhood greenness, environmental psychology, public health, sedentariness, green space, built environment, natural environment, Austria

## Abstract

Previous studies reported contradictory evidence for associations between perceived greenness and obesity mediated by physical activity, focusing on people’s homes or general greenness. Data are lacking in other environments. We studied the association of perceived greenness at home and at university with BMI and physical activity. An online survey collected data from 601 participants, living and studying in and around the city of Graz, Austria; mean age of 24 years. Greenness was assessed using questions on quality of and access to green space; Body mass index (BMI) was derived from self-reported measures; physical activity and sedentariness were measured using the IPAQ questionnaire (short version). On average, BMI was 22.6 (SD = 3.7), physical activity was 63.3 (SD = 51.7) METh/week, and participants spent 5.8 (SD = 4.0) h/day sitting. Regression analyses revealed no associations between perceived greenness and BMI and physical activity for all environments, but a negative association for sedentariness and perceived greenness at home, but not at university. The results indicate a relation between perceived greenness and sedentariness, which differs for the home-and study environment.

## 1. Introduction

Forests, vast grasslands, natural landscapes, and omnipresent greenness—nature has been an integral part of life on earth for a long time. Not until the recent several hundred years in history have industrialization and advancing technology been a result of human evolution. However, these general advancements have also brought grey colored facades, industrial areas, man-made pollution, and serious consequences to health. Urbanization and modern lifestyles as well as obesity and sedentariness are increasing challenges humanity has to face today. Humans have developed and maintained an affinity for nature throughout their evolution. Thus, according to the biophilia hypothesis [1,2], exposure to green spaces provides a buffer against these challenges. Additionally, locations with high degrees of objectively measured [3] or high perceived “neighborhood greenness” [4] can promote physical activity, while reducing obesity [5].

### 1.1. Neighborhood Greenness, Obesity and Physical Activity

General health is affected by personal, socioeconomic, and environmental factors [6]. Further, body-mass-index (BMI) is associated with greenness and the presence of nature [7]. Living closer to green space involves higher odds of leanness in individuals, whereas living further away from green space leads to a higher likelihood of obesity [8]. This may be due to the physical activity-promoting properties of these environments, which support active behavior and thus result in reduced obesity rates [8]. However, there is contradictory evidence in this context—for instance, some studies report positive associations of living close to green areas with physical activity, but no association with obesity [9,10], making the associations less reliable and clear.

Greenness may encourage physical activity due to providing cycling and walking destinations and serving as a place for play and exercise [3]. Indeed, several studies reported moderately positive associations between green space and physical activity in several studies [11,12,13,14,15]. There is evidence that the perception of the living environment can influence physical activity [4] and may be equally predictive of physical activity as objectively measured environmental characteristics [16]. With regard to greenness in the environment, it is not yet clear whether both perceived and objective greenness are associated with physical activity [17,18], and although physical activity may act as a mediator between greenness of the environment and BMI [8], it is likely not the only relevant mediator of this association. A distinction between low levels of physical activity, i.e., “inactivity”, and “sedentariness” needs to be made [19], as research points to distinct health effects of these two concepts [20]. Defined as activities in reclining, seated or lying position accompanied by low energy expenditure [19], sedentary behavior can affect health as an independent predictor, even if physical activity guidelines are met [21]. This makes it an important factor to consider in addition to physical activity levels. However, sedentary behavior has not been studied in relation to greenness of the environment so far.

Previous research was conducted to gather insights into how greenness affects the human body and mind. It was theorized that merely looking at nature is enough to derive positive outcomes on different aspects of mental health [22,23]. In addition, the aesthetic makeup of the immediate environment might be an important factor to initiate the desire to be physically active as supported by a higher likelihood of active behavior in green environments [3,18]. Furthermore, positive effects can be obtained once individuals move or stay in natural environments over a given timeframe [1,2,24], either as a consequence of looking at green space or merely because of its accessibility [4].

### 1.2. Physical Activity and Sedentariness of College Students

College students differ from the average population as their mean age is usually lower [25], they have more leisure time than workers [26], and they are likely to have better education [25]. Reports indicate that the students have physical inactivity rates of about 40% to 50% [27]. Physical activity and sedentary behavior of college students are influenced by social networks, the physical environment, exams or academic pressure [28]. Among men, computer use was identified as a possible driver for sedentary behaviors, and television watching among women. In general, there was more self-reported computer use in older students than in younger ones [29]. These factors may lead to high inactivity among students [27]. Relocation from home to university is another factor associated with decreased physical activity parameters and increased bodyweight in female students [30]. In conclusion, this makes college students, especially women, a subgroup of the general population that is prone to sedentary behavior and poor overall health parameters. As greenness around university campuses was shown to have positive outcomes for mental health markers of students [31], increased physical activity as well as decreased sedentariness and BMI may be expected.

### 1.3. Objective

The existing evidence on why greenness influences physical activity and BMI comes from controlled field studies that compared residents living in a green environment with residents living in a less green environment [32]. However, people do not only spend time at home during the day. Many adults spend much of their time at work, and the environmental properties of the workspace, i.e., greenness and walkability, can also positively influence physical activity levels [33]. Similarly, college students spend much of their time at university or educational facilities. We propose that green around the study place of college students does not have the same effects on physical activity and BMI as green in the home environment. Greenness around the study place may only trigger a desire for subsequent direct greenness exposure [3,18], which is then satisfied in the home environment. In line with human “love” for nature [1], this may be because home environments are rather chosen due to their natural surroundings compared to study environments, where free time is not likely to be spent. This is why we propose that there is no relation between greenness at university, physical activity, and BMI compared to greenness at home, where these associations exist. Furthermore, greenness at the university might be less influenced by socioeconomic status or a person’s own preference for their living environment.

This study extends the current body of evidence by observing the associations between perceived neighborhood greenness, physical activity, sedentariness, and BMI and, specifically, includes perceived greenness at home and greenness at people’s study-places. We hypothesize that there is no relation between greenness at university, physical activity, and BMI, whereas there will be a positive association between greenness at home and physical activity and a negative association with BMI and sedentariness.

## 2. Materials and Methods

The collected data were part of a larger online-survey that was conducted between October and December 2017. Participants with both a residence and study-place in the city of Graz (Austria, capital of Styria) and its proximal areas were selected. Approximately 300,000 residents are living in the city of Graz that spans 128 km^2^, which is predominantly known for its universities. The university area, where the greatest part of the sample lived and/or spent the most time during the day, is surrounded by green space and a park area, which is publicly accessible year-round. In Graz, three main universities of the city (University of Graz, the University of Music and Performing Arts, and the Technical University) have their facilities located in the same area with little distance between their campuses. These universities cooperate with each other and, beyond regular programs, also offer interdisciplinary study programs. Students do not live on campus in Graz, but most have their homes elsewhere in the city of Graz or surrounding areas. Due to the short distance between the university buildings and park area and interuniversity cooperation, Graz was considered a good place to collect data representative for college students in (Mid-)Europe. The ethical committee of the University of Graz approved this study (GZ 39/58/63 ex 2016/17) and participants had to fill in an informed consent prior to taking part in the study.

### 2.1. Study Population and Data Collection

Recruitment of participants was accomplished via university-related mail distribution lists, followed by announcements of the study via social media status updates as well as university-related communities. Therefore, no information is available about the total number of people that were invited to take part in the study. A total of 758 respondents took part in the study, of whom 115 had to be removed due to missing informed consent. Another 42 respondents were excluded from the analyses specifically for the purpose of this study, as their current employment status was different than “college student”. The sample used in this study consisted of 601 participants. We suppose the students of this sample spent most of their time at university indoors, only looking at nature outside, due to the temperature during the months when they participated in the study (October to December). Compared to spring and the summer months, they were less likely to spend prolonged time in green areas on and around the campus, e.g., for learning and passing time between classes.

### 2.2. Study Variables and Questionnaires

#### 2.2.1. Perceived Neighborhood Greenness

Since no standardized questionnaire for perceived neighborhood greenness was available, this measure was obtained using German translations of the following six questions from the PHENOTYPE project [34]. Ranging from 0 to 5 (very good; good; neither good nor bad; bad; very bad), these questions were centered on the quality of green space and the access to it [34,35]. As marked with (1) and (2), the participants had to answer each question two times, first for the environment at home and then for their study place/workplace (or the place where they used to spend the most time during the day), adding up to twelve questions overall:

In your (1) neighborhood, (2) around your study/working place, how would you describe:Access to parks or nature?Access to walking or bicycle paths?Presence of greenness?Presence of trees/tree density or canopy along footpaths?Presence of other natural features?Overall, how would you describe the quality of the green/blue space at your location?

In order to receive two condensed measures for perceived greenness at home and at the participants’ study-place, the answers for each location were averaged and transformed into percentages. Additionally, an overall perceived-greenness variable was used for certain analyses, which was derived from the average of perceived greenness at the home and study environment. This variable was included to make this study more comparable to other studies, which were not making a distinction between greenness in different locations.

#### 2.2.2. Body Mass Index

Participants were asked about their height and bodyweight to obtain BMI scores of each individual (calculated as weight in kg over m^2^). Subsequently, underweight was defined as BMI ≤ 18.5, normal weight as BMI > 18.5 and <25, overweight as BMI ≥ 25, and obesity was defined as BMI ≥ 30 [36]. These definitions were included for interpretation only, as metric measures were maintained throughout the analyses.

#### 2.2.3. Physical Activity and Sedentariness

Physical activity was assessed with the German version of the short form of the IPAQ questionnaire [37]. It asks participants about an estimate of the time per week spent with vigorous, moderate, and light activity, as well as an estimate of the time spent sitting per day without distinction between week or weekend days. The activity measures of the questionnaire were transformed into metabolic equivalents per minute (MET-min) per week, summed up into a single measure for physical activity and transformed into MET–h per week [38]. Outliers (values >400 MET–h per week) were removed. The item on sitting time was transformed into hours per day and used as a measure for sedentariness. This includes the time spent sitting while at work, at home, while doing course work, during leisure time, and driving a motor vehicle on a usual workday [39].

#### 2.2.4. Statistical Analysis

Correlational analyses were conducted via Pearson correlational analyses. They were used to observe the extent to which perceived greenness scores of the home environment correlate with the perceived greenness of the university.

The associations between (1) perceived neighborhood greenness and BMI and (2) perceived neighborhood greenness and physical activity were analyzed using single- and multivariate linear regression models. Perceived neighborhood greenness was included in the analyses for the home environment and the greenness in the neighborhood of the university. Testing for confounders was accomplished by including relevant variables (gender, age, income, education) based on literature [36,37,38,39,40,41]. Missing values in confounders were handled using a multivariate regression model with multiple imputation. Random seed for numbers in the imputation process was set to 1 and the number of imputations to 100 [42]. All Analyses were conducted using IBM SPSS 25^®^, New York, NY, USA.

## 3. Results

### 3.1. Sample

The study sample consisted of 601 students, 465 (77%) of whom were female, with a mean age of 24 (SD = 7) years. Additional characteristics of the sample can be found in Table 1.

#### 3.1.1. Testing for Assumptions of Multivariate Regression Analysis for Confounder Testing

Tests for collinearity were conducted using Pearson correlations for the combined data after multiple imputation. The results can be found in Appendix A. The strongest effect was a moderate correlation between age and income (*r* = 0.52, *p* < 0.001). Tolerance statistics and variance inflation factors in the original dataset indicated that collinearity was not of concern, which can be found in Appendix A. The original data met the assumption of independent errors (Durbin–Watson value BMI = 1.98; physical activity = 1.81; time spent sitting = 1.86) and Cook’s distance values were under a value of 1 for all three imputation models, indicating that individual cases were not unduly influencing the regression model. A scatterplot of standardized residuals in the original dataset and multiple scatterplots for each dataset in the imputation process showed that the data met the assumptions of homogeneity of variance and linearity. Normal P–P plots of standardized residuals showing almost all dots on and close to the line for the original and imputed datasets indicate that the data contained approximately normally distributed errors.

#### 3.1.2. Perceived Neighborhood Greenness at Home and at University

Perceived greenness scores for both the home environment (M = 79.86%, SD = 16.40%) and the study environment (M = 69.78%, SD = 19.45%) were high in general. Additionally, a variable for overall perceived greenness was derived from the average of the previous two measures (*M* = 74.86%, *SD* = 14.14%). Pearson correlations yielded a significant but moderate correlation between perceived greenness at home and at university, *r* = 0.24, *p* < 0.001.

#### 3.1.3. Perceived Neighborhood Greenness and BMI

The results of the analyses of perceived greenness and BMI adjusted for gender, age, income, and education can be found in Table 2 (for an extended version, see Appendix A). Perceived greenness at home, at university, and overall is not associated with BMI. Univariate associations with BMI were similar for greenness at home or at university. When adding greenness of both locations into the model simultaneously, both remained non-significantly associated with BMI. Adjustment for gender, age, income, and education did not change the strength of the associations.

#### 3.1.4. Perceived Neighborhood Greenness, Physical Activity, and Sedentariness

The results of the analyses of perceived greenness and physical activity as well as sedentariness can be found in Table 3 (for an extended version, see Appendix A). Perceived greenness at home, at university, and overall was not associated with physical activity. Univariate associations with physical activity were similar for greenness at home or at university. When adding greenness of both locations into the model simultaneously, both remained non-significantly associated with physical activity. Adjustment for gender, age, income, and education did not change the strength of the associations.

Perceived greenness at home and overall was negatively associated with sedentariness, whereas perceived greenness at university was not. Univariate associations with sedentariness differed for greenness at home and at university. When adding greenness of both locations into the model simultaneously, perceived greenness at home remained significantly associated with sedentariness, and perceived greenness at university remained non-significantly associated with sedentariness.

## 4. Discussion

### 4.1. Main Findings

We did not find associations of perceived neighborhood greenness at home with BMI or physical activity. However, sedentariness was negatively associated with perceived greenness at home. No associations were found between perceived greenness at university and BMI, physical activity or sedentariness.

Contrary to our hypothesis and many previous studies [4,17,18,32], we did not find an association of perceived greenness at home with higher levels of physical activity or a lower BMI. Differences might be explained by our relatively lean and young study population. However, our population was representative of the total population of college students, and in line with previous studies’ BMI distributions and low physical activity levels in college students [27,43]. Another reason could be a possible non-linear relationship between greenness and physical activity [44]. Different study designs would have been necessary to detect these associations. However, as expected, we found a negative association between perceived greenness at home and sedentariness, which indicates that a greener home environment can lead to a somewhat higher energy expenditure. The estimated change in sedentary time with increased perceived greenness is only small and likely not relevant at the societal level. However, this association has not been studied before, and our finding warrants further investigation, preferably in future studies with a more representative study sample and objective measures of physical activity and sedentary behavior.

The finding of perceived greenness of the study environment not being associated with BMI, physical activity or sedentariness was according to our hypothesis. For an effect on physical activity or BMI, being physically present in the green area is needed, and it is likely that for most students, this kind of direct exposure is not possible during the time they spend at university. Further, they might prefer being active in other settings, such as a fitness or sports club [45]. However, more information on what students do during breaks and how they use green spaces in the environment would be of use to further understand the (lack of) relationship between greenness and physical activity and BMI.

Why there were associations between perceived greenness at home and sedentariness, but not for university, might also be explained by the biophilia hypothesis [1,2]: According to human preference for nature [46,47,48], residents prefer to visit green spaces over a built environment in the neighborhood of their homes. We assumed that the study-place is not likely to be chosen because of its environmental properties, and in this context, also not for leisure time activities.

Moreover, temporal and spatial dimensions need to be taken into consideration in regard to when sedentariness is less likely to occur, e.g., during free time and with certain modes of transport. Study environments might allow for activity to a limited extent only compared to living environments, where most of the free time is spent. Accordingly, nature in the neighborhood of people’s homes, access to parks or bikeways, and sidewalks in and around green spaces promote physical activity [3]. Based on our results that increasing rates of perceived neighborhood greenness at home are associated with decreasing time spent sedentary, we conclude that time might rather be spent being active, even if no significant results could be obtained for direct activity measures in this study due to invalid measurements or other reasons.

### 4.2. Mechanisms of Action

The greatest part of the prevailing literature on green space, physical activity, and obesity did not incorporate greenness in distinct locations [32]. Thus, including the perceived greenness measures at people’s living environments and study-places reveals new insights into how nature affects these markers of human health. Due to this distinction, we were able to broaden the existing research by revealing different effects of perceived greenness in different locations on physical activity and sedentariness. As people are not likely to only spend time at home, it needs to be considered that neighborhood greenness should be assessed for different places. It is unclear whether nature acts synergistically through both direct exposure and as a motivator for physical activity [3,18] or whether greenness in one distinct location has the greatest impact on health. Certainly, future research should address these modalities in closer detail in order to see if distinct effects regarding BMI and physical activity can be obtained for home, study, work, and other environments where people spend most time of the day, while incorporating more heterogeneous samples.

Although the literature predominantly reports that perceived neighborhood greenness affects physical activity [4], the relationship between these factors could simultaneously act in a reverse manner: Active individuals might perceive their home environment to be greener or qualitatively better than less active individuals, who do not leave their homes for physical activity alone or within a social circle. We suppose that active individuals are likely to develop a stronger relationship to their neighborhoods, including their physical and social environment, which could enhance their perception of the quality of their immediate environment. There is research showing the importance of social participation for community identity and that intercommunity spatial activity promotes a regional identity [49]. We believe that this identity, as a sense of home, could positively influence the perceived living environment, which might be another reason this study could not detect any significant associations between perceived greenness at university and physical activity as well as sedentariness.

### 4.3. Methodological Considerations

Perceived greenness at home and at university was relatively high. We suppose this is due to the generally green campus of the University of Graz and the small distance to park areas around it. We checked for a possible subjective bias, with some people perceiving the environment to be greener than others. The perceived greenness at home and at university had only a moderate correlation, indicating that there might not be a strong influence on the outcomes of the analyses from “general high and low greenness-perceivers”.

Due to the online data assessment, there was no possibility to verify BMI scores via objective measurements. There can be considerable error between subjectively measured BMI and real scores, especially for women [50]. Hence, some bias in self-reported BMI must be considered. Although a larger error can be expected for women [50], it is not clear whether there are other systematic or unsystematic errors in the sample of this study, possibly influencing the results of the analyses with BMI as an outcome.

The current results should only be interpreted in the context of sample size due to possible influences on the observed statistical significances. There are predominantly low to moderate effect sizes reported from previous research on neighborhood greenness and obesity [9], which is congruent with the findings of this study. Since all of the effect sizes were rather small in combination with a large sample, a certain limitation of the “clinical” relevance of the results must be taken into consideration. However, the effects of the findings can still be interpreted, since they are similar to other studies addressing similar objectives [51]. It should further be noted that the homogeneity of the student sample in regard to the perceived greenness levels at the university’s environment might be one reason for the small effect sizes in our study. As the majority of the participants lived near and studied in the same places, a limited variance of these greenness scores must be taken into consideration. This implies that larger effects may be possible for more diverse samples.

Since more favorable residential environments in cities, i.e., highly green environments, are rather chosen by people who can afford to live there [52], socioeconomic status and income are expected to exert an influence on the outcomes of studies not controlling for these factors. This student sample is also homogeneous in respect of age, socioeconomic status, income, and education. Due to this makeup of the sample, the majority had a high education and an income lower than 1000 € per month. This means that confounding by socioeconomic factors was unlikely to occur. However, it also means that the results cannot be generalized to all adult residents in Graz.

No standardized and validated questionnaires for perceived greenness are currently available. Thus, it is not clear if the questions assessing perceived greenness measures were reliably measuring the desired outcomes. The subset of questions we took from the PHENOTYPE questionnaire [35] can be compared to items from the Neighborhood Environment Walkability Scale [53]. This scale was already used in a similar study focusing on perceived greenness [54] and contains items concerned with access to park or nature reserve, access to bicycle and walking paths, presence of greenery, and presence of tree cover or canopy along footpaths, as well as presence of pleasant natural features [53]. Although this does not influence the reliability of the questions used in the study at hand, it increases the comparability of the perceived greenness measures of other studies.

Similar to most previous studies, the study at hand is of a cross-sectional nature, which implies that the direction of the results in terms of causation is not clear, which makes a causal relationship between greenness and sedentariness impossible to establish.

### 4.4. Prospect

An important determinant of BMI is energy intake, i.e., nutrition and eating behavior, which could counteract a high energy expenditure from vigorous physical activity and thus diminish the effects of the environment on BMI over time. There is some support for this assumption by a study observing the relation between obesity and environmental variables. It revealed that people’s home environment has an influence on their eating behavior in a way that areas with obesity-promoting diets in children were characterized by smaller streets, busy roads, little open space, and many food outlets, compared to less obesity-promoting diets in areas with open space, wider streets, greenness, and fewer food outlets [55]. Therefore, it would be useful to take eating behavior into account in future studies on the relationship between greenness in the neighborhood and BMI.

Future research could also have a closer look at the association between both perceived and objective greenness and obesity mediated by physical activity in the context of different environments such as at home, at university, and at work in order to see if more pronounced effects can be obtained. As we tried to demonstrate, different locations may vary in the effects of direct and indirect exposure to nature. More research is needed to explore the implications of this new approach. In line with this, more valid and reliable methods are needed to approach the relationships between physical activity and greenness in similar study designs. Including measures for objective greenness and objectively assessed physical activity via activity meters in a study design would be valuable for increased reliability of the results for the perceived environmental measures and self-reported physical activity and sedentariness.

## 5. Conclusions

This study among college students found negative associations between perceived greenness at home and sedentariness, but not for the environment at university. Future studies are needed that incorporate more valid and reliable methods in more diverse populations.

## Figures and Tables

**Table 1 ijerph-17-00235-t001:** Sample characteristics.

Variable	Value(s)
	*N* = 601
Gender	465 (77%) female
	125 (21%) male
	11 (3%) missing
Marital status	20 (3%) married
	258 (43%) in a partnership
	281 (47%) single
	42 (7%) missing
Income per month	374 (62%) less than 1000 €
	88 (15%) between 1001 € and 3000 €
	10 (2%) more than 3001 €
	129 (22%) missing
Living in the city of Graz	450 (75%)
Studying in the city of Graz	543 (90%)
Age (years), *M* ± *SD*	24 ± 7
BMI (kg/m^2^), *M* ± *SD*	22.55 ± 3.69
BMI Category	31 (5%) obese
	78 (13%) overweight
	443 (74%) normal-weight
	38 (6%) underweight
	11 (2%) missing
Physical activity (total MET-h per week), *M* ± *SD*	62.72 ± 51.71
Time spent sitting (hours/day), *M* ± *SD*	5.75 ± 3.95
Perceived greenness at home (%), *M* ± *SD*	79.86% ± 16.40%
Perceived greenness at university (%), *M* ± *SD*	69.78% ± 19.45%
Perceived greenness overall (%), *M* ± *SD*	74.86% ± 14.14%

**Table 2 ijerph-17-00235-t002:** Linear regression models with BMI as outcome and perceived greenness as well as possible confounders as indicators.

Analyses	*R* ^2^	*b*	95% CI *b*	*p*
BMI				
At home	<0.001	0.01	−0.02 to 0.02	0.966
At university	0.001	0.01	−0.01 to 0.02	0.449
Overall	0.001	0.01	−0.02 to 0.03	0.579
Multivariate				
At home and at university	0.001			0.746
At home		−0.001	−0.02 to 0.02	0.906
At university		0.01	−0.01 to 0.02	0.446
Multivariate adjusted				
With confounders				
At home		−0.01	−0.03 to 0.01	0.518
At university		0.01	0.01 to 0.03	0.361
Gender		0.65	−0.07 to 1.38	0.077
Age		0.15	0.10 to 0.21	<0.001
Income		−0.19	−1.06 to 0.67	0.660
Education		−0.10	−0.44 to 0.24	0.563

**Table 3 ijerph-17-00235-t003:** Linear regression models with physical activity and sedentariness as outcomes and perceived greenness measures as indicators.

Physical Activity (Met-h Per Week)				
Analyses	*R* ^2^	*b*	95% CI *b*	*p*
Univariate				
At home	0.002	0.14	−0.15 to 0.44	0.344
At university	<0.001	0.01	−0.23 to 0.26	0.919
Overall	0.001	0.11	−0.24 to 0.45	0.536
Multivariate				
At home and at university	0.003			0.586
At home		0.16	−0.14 to 0.46	0.304
At university		−0.02	−0.27 to 0.24	0.897
Multivariate adjusted				
With confounders				
At home		0.23	−0.08 to 0.54	0.147
At university		−0.03	−0.30 to 0.24	0.809
Gender		−5.66	−17.59 to 6.26	0.352
Age		−0.28	−1.13 to 0.57	0.519
Income		3.52	−8.73 to 15.77	0.573
Education		−5.59	−11.18 to −0.001	0.050
**Sedentariness (Time Spent Sitting, Hours per Day)**				
**Analyses**	***R*^2^**	***b***	**95% CI *b***	***p***
Univariate				
At home	0.02	−0.03	−0.05 to −0.01	**0.003**
At university	0.01	−0.01	−0.03 to 0.003	0.123
Overall	0.02	−0.03	−0.05 to −0.01	**0.006**
Multivariate				
At home and at university	0.02			**0.009**
At home		−0.03	−0.44 to −0.01	**0.008**
At university		−0.01	−0.02 to 0.01	0.532
Multivariate adjusted				
With confounders				
At home		−0.04	−0.06 to −0.01	**0.010**
At university		−0.01	−0.03 to 0.02	0.487
Gender		0.22	−0.57 to 1.01	0.587
Age		0.06	−0.01 to 0.12	0.076
Income		−0.60	−1.50 to 0.30	0.192
Education		0.21	−0.16 to 0.58	0.272

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
