# Peer review of "Sedentariness of College Students Is Negatively Associated with Perceived Neighborhood Greenness at Home, but Not at University"

_ijerph, 2019, doi:10.3390/ijerph17010235_

Round 1
Reviewer 1 Report
Comments to authors
I had the opportunity to review your paper about “Sedentariness is negatively associated with Perceived Neighborhood Greenness at Home, but not at Work / University.” The authors examined the association of perceived greenness at home and work/university with BMI and physical activity. It is very interesting work and has some merits. The structure of the paper was well organized. However, I have some questions.
-Title: because the majority of the respondents are students/young adults, it may be more appropriate to modify the title to fit the results/respondents’ characteristics.
-p2. You stated that the online-survey questionnaire was distributed to respondents via social media. Did you approach the respondent one by one via their messenger? Or, you just sent to some of the respondents and then asked them to share the survey link among their friends?
-p3. I think most of the respondents (94%) are from the University of Graz where you also belong, so it is important to briefly introduce the university’s environment, i.e., green space/parks, and why you chose it. The mean age is 24 (p4), which is likely at the undergraduate level. If so, most of their time may be in class because in general undergraduate students have many courses to complete. Furthermore, I am not sure how the student life at your university is, but for example in China, most of the students are staying in dorms, not at home, and they spend most of the time in classes and libraries. As such, no matter how perceived neighborhood greenness is, sedentariness is inevitably longer.
-p4. Table 1: the percentage (%) for some statistics were missing.
-p4. About 88% of the respondents were single and in a partnership. Some studies emphasized that this group of people prefers more active leisure, such as club or bar entertainments, to wilderness or parks (Chen & Jim, 2008). Did you consider this factor?
-p4. For physical activity, did you include or measure the distance from home or work/university to the green areas/parks? How about the accessibility and functionality of the green spaces/parks? Some studies indicate that the safety and accessibility of green spaces play a very important role in physical activity (Yen et al., 2017). It is also important to take into account the time break. The shortest time break after class or work may limit their physical activity. Do you also know the main activities on campuses that the students generally do? This may help you understand why they are not physically active in green spaces.
-p5. In general, obesity rates are the lowest among adults age 18 to 24 and the highest among those ages 45 and 74 (www.wihealthatlas.org). Your study also showed that 73% of the respondents had a normal weight. Therefore, the association between perceived greenness and BMI may not be significant.
-The main finding is a negative association between perceived greenness at home and sedentariness. It is likely a small finding. How can you generalize this finding?
In short, the paper addresses the gap in the area of interest, but it requires the authors to make the findings more generalizable.
Reference
Chen, W. Y., & Jim, C. Y. (2008). Cost–benefit analysis of the leisure value of urban greening in the new Chinese city of Zhuhai. Cities, 25(5), 298-309. doi:https://doi.org/10.1016/j.cities.2008.06.002
Yen, Y., Wang, Z., Shi, Y., Xu, F., Soeung, B., Sohail, M. T., . . . Juma, S. A. (2017). The predictors of the behavioural intention to the use of urban green spaces: The perspectives of young residents in Phnom Penh, Cambodia. Habitat International, 64, 98-108. doi:10.1016/j.habitatint.2017.04.009
Author Response
-Title: because the majority of the respondents are students/young adults, it may be more appropriate to modify the title to fit the results/respondents’ characteristics.
We changed the title as suggested. Furthermore, based on your suggestion and comments from other reviewers, we limited the whole paper to students by excluding 42 participants who were not a student.
-p2. You stated that the online-survey questionnaire was distributed to respondents via social media. Did you approach the respondent one by one via their messenger? Or, you just sent to some of the respondents and then asked them to share the survey link among their friends?
We did not use a one-by-one approach. We assume many participants were recruited via the mail distribution lists of the university, as many online questionnaires were filled out during the first few days after sending the e-mails. After these initial days, we further announced the existence of the study via status updates and posts in university-related communities on social media, but did not receive the same increase in response from this strategy. We adjusted the sentence on how we approached participants via social media: “Recruiting of participants was accomplished via university-related mail distribution lists, followed by announcements of the study via social media status updates as well as university-related communities.”
-p3. I think most of the respondents (94%) are from the University of Graz where you also belong, so it is important to briefly introduce the university’s environment, i.e., green space/parks, and why you chose it. The mean age is 24 (p4), which is likely at the undergraduate level. If so, most of their time may be in class because in general undergraduate students have many courses to complete. Furthermore, I am not sure how the student life at your university is, but for example in China, most of the students are staying in dorms, not at home, and they spend most of the time in classes and libraries. As such, no matter how perceived neighborhood greenness is, sedentariness is inevitably longer.
Thank you for this important suggestion, and we agree that it is important to make the context of the study clearer. In the first paragraph of the Methods section, we now describe the university area in closer detail and stated, why Graz was a good choice for data collection. In the second paragraph of the Methods section we address student life a bit more in detail.
-p4. Table 1: the percentage (%) for some statistics were missing.
We believe this might have been for variables that were not categorical. We provided the mean and SDbut in a similar format as the nand (5) for categorical variables. We have made the distinction between categorical and numerical variables clearer in the table. Please note that some of the variables are measured as a percentage of the total score, and therefore the mean and SDare also expressed in %.
-p4. About 88% of the respondents were single and in a partnership. Some studies emphasized that this group of people prefers more active leisure, such as club or bar entertainments, to wilderness or parks (Chen & Jim, 2008). Did you consider this factor?
We agree that some young people might prefer other ways and places for being active than being outdoors in green. Although an interesting and important aspect, we could not consider this factor, since we did not collect data on this.
-p4. For physical activity, did you include or measure the distance from home or work/university to the green areas/parks? How about the accessibility and functionality of the green spaces/parks? Some studies indicate that the safety and accessibility of green spaces play a very important role in physical activity (Yen et al., 2017). It is also important to take into account the time break. The shortest time break after class or work may limit their physical activity. Do you also know the main activities on campuses that the students generally do? This may help you understand why they are not physically active in green spaces.
In this paper, we did not use objective information on greenness but only the perception of green in the environment. However, anecdotally, we know that many students have easy access to a park when they are at university. Information on what they do during breaks is lacking, as well as information on HOW green was used. We do agree with the reviewer that those aspects are important, but unfortunately in this study we could not collect this type of data. We have added the importance of this information to the discussion.
-p5. In general, obesity rates are the lowest among adults age 18 to 24 and the highest among those ages 45 and 74 (www.wihealthatlas.org). Your study also showed that 73% of the respondents had a normal weight. Therefore, the association between perceived greenness and BMI may not be significant.
We agree with the reviewer, and actually mentioned this aspect in our discussion (page 8).
-The main finding is a negative association between perceived greenness at home and sedentariness. It is likely a small finding. How can you generalize this finding?
We agree that the association between perceived greenness at home and sedentary time is weak, with only a small decrease in sitting time with increased greenness. However, this is the first study assessing this association and we cannot rule out that this association might be larger in other populations. Therefore, a future study with a more representative study sample, and with objective measures of physical activity and sedentary time would be needed. We have added these considerations to the discussion.
Reviewer 2 Report
This paper presents to demonstrate the associate of perceived greenness at home and at work / university with sedentariness.
I have found a few issues that, once addressed, will improve this paper.
P.2.L.20.
(error)green spaces in in the neighborhood (correct)green spaces in the neighborhood
P.3.
Physical Activity
Please explain Sedentariness in detail.
Because,
Sedentariness is an important variable related to your conclusion. Sedentariness is an indefinitely difficult variable to measure and evaluate in IPAQ questionnaire (short version). Sedentariness is rarely analyzed in other papers, and it is difficult to compare the evaluation of this paper with other papers.
P4. L2.
(error)All Analyses (correct)All analyses
Please provide the IBM SPSS 25 company location.
All Tables.
If there is a significant difference in the p-values in the tables, emphasis using bold letters makes it easier to see the tables.
Discussion
I would like you to enter the limit that there is a considerable error between the reported BMI and the measured BMI, especially for women.
For example,
“Validity of Self-Reports of Height and Weight among the General Adult Population in Japan: Finding from National Household Surveys, 1986”. Plos One, 11: e0148297, 2016.
etc.
In this paper, all variables are based on “subjective” data.
In the next paper, I would like to analyze at least one, for example, physical activity, based on “objective” data using “activity meter”.
Because, it is inaccurate to look back on human activities later and fill out the questionnaire, and the results are likely to be distorted.
I hope these comments will be helpful.
Author Response
P.2.L.20.: (error)green spaces in in the neighborhood (correct)green spaces in the neighborhood
We deleted the additional “in”.
P.3.: Physical Activity. Please explain Sedentariness in detail. Because, sedentariness is an important variable related to your conclusion. Sedentariness is an indefinitely difficult variable to measure and evaluate in IPAQ questionnaire (short version). Sedentariness is rarely analyzed in other papers, and it is difficult to compare the evaluation of this paper with other papers.
Sedentariness was measured with the IPAQ. Participants were asked to estimate the time they spent sitting per day. This includes the time spent sitting while at work, at home, while doing course work, during leisure time and driving a motor vehicle on a usual workday.
P4. L2.: (error)All Analyses (correct)All analyses
We corrected this as suggested.
Please provide the IBM SPSS 25 company location.
In the section “Statistical Analyses” we included the location (New York, USA).
All Tables: If there is a significant difference in the p-values in the tables, emphasis using bold letters makes it easier to see the tables.
We used bold font for significant p-values < .05 as suggested.
Discussion
I would like you to enter the limit that there is a considerable error between the reported BMI and the measured BMI, especially for women. For example, “Validity of Self-Reports of Height and Weight among the General Adult Population in Japan: Finding from National Household Surveys, 1986”. Plos One, 11: e0148297, 2016. etc.
Thank you for this suggestion. This is a good point, which is very relevant in the context of our data. We included a short paragraph on this point in the methodological considerations of the discussion.
In this paper, all variables are based on “subjective” data. In the next paper, I would like to analyze at least one, for example, physical activity, based on “objective” data using “activity meter”. Because, it is inaccurate to look back on human activities later and fill out the questionnaire, and the results are likely to be distorted.
We agree that objective measurement of physical activity and/or BMI would be ideal, since this would reduce bias. We have added this in the discussion as a need for future studies (Prospect).
Reviewer 3 Report
General
The manuscript focuses on exploring the association between sedentary behavior and perceived greenness at home/work/ study, which mainly uses perceived data of the online survey. However, there are a number of issues that the authors will need to reexamine.
1 Study design
The main problem is that nearly all the data(94%) come from college students , which should be treated carefully throughout the whole paper, especially in scientific question definition and result interpretation.
The title \ abstract\ research design should focus on the characteristic of physical activities and the routine life of college students.
The physical environment around campus also should be discussed. It would be better the accessibility of green space was added as a comparison.
2 This paper did not conclude enough recent references
most of them were before 2015, which also limited the research gap defined by this paper. At least, the following references are highly related to this manuscript.
1)Association between cardiometabolic risk factors, physical activity and sedentariness in Chilean university students,2017, Europe PMC
2)Associations Between Worksite Walkability, Greenness, and Physical Activity Around Work,2018, Environment and behavior
3)Sedentariness and Health: Is Sedentary Behavior More Than Just Physical Inactivity? 2018, Frontiers in Public Health
4)GPS-Based Exposure to Greenness and Walkability and Accelerometry-Based Physical Activity, 2017, Cancer epidemiology.Biomarkers and Prevention
3 The whole paper needs to be checked carefully, such as
1) missing reference
However, there is contradictory evidence in this context, for instance, reporting positive associations between living close to green areas and physical activity, but no outcomes on obesity [8,9](paragraph 2 in Introduction)
reference 9 is missing
2) inappropriate reference
The author used a manuscript under review as a reference.
31 Loder, A.K.F.; Schwerdtfeger, A.; Van Poppel, M.N.M. Perceived Greenness at Home and at Work / University are independently Associated with Mental Health. BMC Public Health 2019 (submitted). * This paper is currently under review.
3) confusing statement
is the "Perceived" wrong, according to the context
There is evidence that the perceived living-environment can influence physical activity, for instance when parks are accessible in the neighborhood [4]. (paragraph 3 in Introduction)
4)exaggerated statement
the manuscript seem unable to provide insight into the mechanism of "How"
Research was conducted to gather insights into how greenness affects the human body and mind.(paragraph 4 in Introduction)
Author Response
1 Study design
The main problem is that nearly all the data (94%) come from college students,which should be treated carefully throughout the whole paper, especially in scientific question definition and result interpretation.
The title \ abstract\ research design should focus on the characteristic of physical activities and the routine life of college students.
The physical environment around campus also should be discussed. It would be better the accessibility of green space was added as a comparison.
Based on your comment and those of other reviewers, we concluded that it would be better for our paper to use an exclusive student sample. We reanalyzed the data including only participants who answered the question “Are you a student?” with “yes”.
Furthermore, we concentrated on the specific characteristics of this sample and excluded the workplace. We have provided more details on the study place and the campus environment. We also addressed the focus on students in several paragraphs throughout the paper and adjusted / extended these sections accordingly.
2 This paper did not conclude enough recent references
most of them were before 2015, which also limited the research gap defined by this paper. At least, the following references are highly related to this manuscript.
1)Association between cardiometabolic risk factors, physical activity and sedentariness in Chilean university students,2017, Europe PMC
2)Associations Between Worksite Walkability, Greenness, and Physical Activity Around Work,2018, Environment and behavior
3)Sedentariness and Health: Is Sedentary Behavior More Than Just Physical Inactivity? 2018, Frontiers in Public Health
4)GPS-Based Exposure to Greenness and Walkability and Accelerometry-Based Physical Activity, 2017, Cancer epidemiology.Biomarkers and Prevention
We added these additional references to the introduction and discussion.
3 The whole paper needs to be checked carefully, such as
1) missing reference
However, there is contradictory evidence in this context, for instance, reporting positive associations between living close to green areas and physical activity, but no outcomes on obesity [8,9](paragraph 2 in Introduction). Reference 9 is missing
Thank you making us aware of this point. We have checked the reference list again and corrected any mistakes.
2) inappropriate reference
The author used a manuscript under review as a reference.
31 Loder, A.K.F.; Schwerdtfeger, A.; Van Poppel, M.N.M. Perceived Greenness at Home and at Work / University are independently Associated with Mental Health. BMC Public Health 2019 (submitted). * This paper is currently under review.
We have deleted this reference from the manuscript, and changed the text in the discussion slightly because of this.
3) confusing statement
is the "Perceived" wrong, according to the context. There is evidence that the perceived living-environment can influence physical activity, for instance when parks are accessible in the neighborhood [4]. (paragraph 3 in Introduction)
We left out the second part of this sentence as the previous sentences already go into detail with objective measures. This makes it clearer and there might be no need for another example.
4) exaggerated statement
the manuscript seem unable to provide insight into the mechanism of "How". Research was conducted to gather insights into how greenness affects the human body and mind.(paragraph 4 in Introduction)
This sentence referred to previous studies on this topic and not our own study. We have made this clearer in the text.
Round 2
Reviewer 1 Report
Comments to authors
I could see a good effort of the authors in revising the manuscript. Basically, it is ok; however, I think that literature (introduction) is still weak. Since the majority of your respondents were female and the mean age was 24, the authors should add some literature about this group. It is thus suggested that the authors need to strengthen the literature to meet the standard of IJERPH.
Author Response
I could see a good effort of the authors in revising the manuscript. Basically, it is ok; however, I think that literature (introduction) is still weak. Since the majority of your respondents were female and the mean age was 24, the authors should add some literature about this group. It is thus suggested that the authors need to strengthen the literature to meet the standard of IJERPH.
We extended the introduction describing our specific subsample in relation to our major outcome variables.
Reviewer 3 Report
The overall quality of this manuscript has been largely improved, but this paper still needs to clarify the research gap in terms of college students.
(1) The authors still need to define the features of daily life activities of college students which would affect the time of sedentariness and the value of BMI.
In the introduction, the author should provide more background knowledge and answer what is the difference between college students and others , such as colleges students may have more leisure time than ordinary workers, or have some sports courses, or may prefer using gym, and they usually have better education or healthier BMI than ordinary people, and etc.
then, the discussion will compare the result with the previous studies about the sedentariness of college students, not standard people.
(2)The title should highlight the special group of college students.
(3) some background knowledge may need to be added too,such as the difference in BMI as a personal health index, which may also be affected by personal state, socioeconomic status, and environmental exposure indicated by the following reference.
Maternal Health and Green Spaces in China: A Longitudinal Analysis of MMR Based on Spatial Panel Model, Healthcare, 2019
Author Response
The overall quality of this manuscript has been largely improved, but this paper still needs to clarify the research gap in terms of college students.
(1) The authors still need to define the features of daily life activities of college students which would affect the time of sedentariness and the value of BMI.
In the introduction, the author should provide more background knowledge and answer what is the difference between college students and others, such as colleges students may have more leisure time than ordinary workers, or have some sports courses, or may prefer using gym, and they usually have better education or healthier BMI than ordinary people, and etc.
then, the discussion will compare the result with the previous studies about the sedentariness of college students, not standard people.
We extended the introduction and discussion describing our specific subsample in relation to our major outcome variables.
(2)The title should highlight the special group of college students.
We changed the title and included the specific sample.
(3) some background knowledge may need to be added too, such as the difference in BMI as a personal health index, which may also be affected by personal state, socioeconomic status, and environmental exposure indicated by the following reference.
Maternal Health and Green Spaces in China: A Longitudinal Analysis of MMR Based on Spatial Panel Model, Healthcare, 2019
We included this additional information in the introduction and added the reference.